# Ramadan During Pregnancy and Offspring Age at Menarche in Indonesia: A Quasi-Experimental Study

**DOI:** 10.3390/nu17091406

**Published:** 2025-04-22

**Authors:** Van My Tran, Reyn van Ewijk, Fabienne Pradella

**Affiliations:** 1Chair of Statistics and Econometrics, Faculty of Law, Management and Economics, Johannes Gutenberg-University Mainz, Jakob-Welder-Weg 4, D-55128 Mainz, Germany; vanewijk@uni-mainz.de (R.v.E.); fapradel@uni-mainz.de (F.P.); 2Heidelberg Institute of Global Health (HIGH), Heidelberg University Hospital, D-69120 Heidelberg, Germany; 3Division of Primary Care and Population Health, Department of Medicine, Stanford University, Stanford, CA 94305, USA

**Keywords:** Ramadan, fetal programming, maternal nutrition, Indonesia, menarche, reproductive health, quasi-experiment, nutrition in pregnancy, intermittent fasting, Muslim

## Abstract

**Background/Objectives**: Animal models have suggested a link between maternal nutrition and offspring pubertal onset. Due to ethical and practical concerns, human studies on this topic remained scarce and focused on extreme nutritional shocks in high-income settings, such as Dutch famine. This paper expands on these findings by investigating the effects of a milder form of nutritional alteration during pregnancy—Ramadan fasting—in a middle-income context, Indonesia. We use offspring age at menarche (AAM) as an indicator of pubertal timing and female reproductive health. Our research has broader implications beyond the Muslim community, as intermittent fasting during pregnancy is also widely practiced by non-Muslims, e.g., meal-skipping. **Methods**: We used data from the Indonesian Family Life Survey (1993–2014, n = 8081) and Indonesian Demographic and Health Surveys (2002–2007, n = 13,241). OLS and Cox regressions were applied to compare the AAM of female Muslims who were prenatally exposed to Ramadan and those of female Muslims who were not. Exposure was determined based on the overlap between pregnancy and a Ramadan. We further subdivided this overlap into trimester-specific categories, adjusting for urban–rural residence, birth month, birth year, birth year squared, and survey wave. **Results**: No associations between Ramadan during pregnancy and AAM were found, irrespective of the pregnancy trimester overlapping with Ramadan. These results were stable when we restricted the sample to women with shorter recall periods and younger women at the time of survey. **Conclusions**: While subtle restrictions in maternal nutrition during pregnancy are critical for offspring health, the impact on menarcheal onset might be limited.

## 1. Introduction

The age at menarche (AAM), a woman’s first menstrual cycle, is an important indicator of female reproductive health [1,2]. The AAM predicts key indicators of fertility such as fecundity [3], birth rate [4], and age at menopause [5]. Furthermore, early menarche was associated with increased risks of breast cancer, cardiovascular diseases, and mortality [6,7,8], while delayed menarche was linked to osteoporosis and reduced areal bone density [6]. Even though genetics is an important determinant, environmental factors may also affect menarcheal timing [6]. Given the limited possibility to influence genetic aspects, illuminating the mechanisms that connect non-genetic factors with the AAM could provide crucial insights for mitigating the associated reproductive and general health consequences.

A major non-genetic determinant of the AAM is nutrition. While the evidence on the link between early childhood or adolescent diets and the AAM has been well established [9,10,11], research on the impacts of maternal nutrition during pregnancy on offspring AAM remains limited. Prenatal nutrition may influence the AAM through dynamic changes in the hypothalamic–pituitary–gonadal (HPG) axis, which regulates reproductive hormone production [12,13,14]. Such an alteration is explained by fetal programming theory, which predicts that environmental factors in utero can cause long-term functional and structural changes in organisms [15]. This hypothesis has been indicated in studies on sheep and lambs, where changes in releasing hormonal levels within the HPG axis were shown to be triggered by maternal undernutrition, specifically during early pregnancy [16,17]. Moreover, experimental rat models suggested that maternal diet alterations during gestation may affect offspring pubertal onset [18,19,20,21,22]. Genetic studies in humans also support this finding, showing that DNA methylation, a determinant for gene expression within the HPG axis, is particularly sensitive to environmental factors during early gestation [23].

In human studies, the existing evidence on prenatal nutrition and the AAM is both scarce and inconsistent. Due to ethical concerns, it is infeasible to assign nutritional restrictions to pregnant women. Furthermore, controlled–randomized trials would require long follow-up periods as menarcheal timing is observable only during adolescence. Thus, the available studies are limited with regard to study designs. An Indian study using data from a supplementary nutrition program reported later menarche in the offspring of pregnant women who had received a balanced protein–calorie supplement during pregnancy [24]. By contrast, the Dutch famine, a historical severe prenatal nutritional shock, was not found to be associated with the AAM [25,26]. Research in which birthweight was used as a proxy for nutritional status during gestation reported mixed results, with both higher and lower birth weights reportedly being associated with an earlier AAM [9,11]. Considering the importance of menarche for female reproductive health, evidence from additional settings can yield important new insights into the prenatal nutrition–offspring AAM nexus.

Our Indonesian study employed a quasi-experimental design, similar to that used in the Dutch famine studies; however, instead of prenatal exposure to famine, we used Ramadan during pregnancy as a proxy for maternal malnutrition to assess its impact on offspring AAM. During Ramadan, adult Muslims abstain from food and drink from sunrise to sunset, and in utero exposure to this fasting period can be considered a form of maternal nutritional shock. In contrast to the extreme nutritional shocks induced by famines, Ramadan fasting can be considered a relatively mild nutritional exposure—comparable to meal skipping, a common practice among pregnant women. Meal skipping has been documented across Asia, the Americas, and Europe [27,28,29,30,31,32]. Therefore, the relevance of this research for female offspring extends beyond the context of Ramadan itself.

Even though pregnant women can skip fasting during Ramadan by compensating for it at a later point of time or making expiatory payments to feed the poor [33,34], surveys from various countries showed that between 45 and 99% decide to fast, depending on the cultural context [35,36,37,38,39,40,41,42]. The estimated fasting rate among pregnant Muslims in Indonesia is 68–82% [34,39,43]. Ramadan during pregnancy has been found to be associated with various adverse health and human capital outcomes in the offspring [40], including childhood growth [44]. Such anthropometric outcomes are not only important predictors of AAM [11,13,45], but are also correlated with obesity, an important risk factor of an earlier onset of menarche [46,47]. For pregnant female adults, dietary adaptations to Ramadan might lead to fluctuations in the levels of key reproductive biomarkers such as follicle-stimulating hormone, estrogen, progesterone, and leptin [48]. While initial analyses in our previous research did not identify the AAM as a mechanism linking prenatal nutritional deprivation and offspring height growth, we believe that this potential link warrants further investigation with more rigorously defined, younger and larger samples to overcome recall bias related to the self-reported AAM [49].

A large share of pregnant Muslims fast during Ramadan [35,36,37,38,39,40,41,42], and Muslim births constitute about 31% of the 130–140 million babies born worldwide each year [50,51]. Thus, the findings of this study may hold particular relevance for Muslims of childbearing age and their healthcare providers.

## 2. Methods

### 2.1. Data

Our data come from Indonesia, the country with the world’s largest Muslim population. They comprised two sources: the Indonesian Family Life Survey (IFLS), and the Indonesian Demographic and Health Survey (DHS). IFLS is a longitudinal socioeconomic and health survey that is representative of 83% of the Indonesian population (excluding a few provinces). Its sampling frame was stratified in provinces, from which the households were selected randomly for face-to-face interviews [52]. We used data from all five IFLS waves, conducted in 1993/94, 1997/98, 2000, 2007/08, and 2014/15.

The Indonesian DHS is a nationally representative household survey, utilizing systematic sampling in multiple stages to select individuals for interviews [53]. This sampling was stratified at both national and provincial levels. The surveys were conducted every five years, and data on AAM come from the module of Young Adult Reproductive Health, available in 2002, 2007, 2012, and 2017.

The IFLS sample includes 8081 ever-married women aged 18 to 60, born between 1935 and 1997. The DHS sample consists of 24,898 never-married Muslim women aged 15 to 25, born between 1978 and 1999, with the main analysis sample being restricted to 13,104 women at the age of 18 to 25. We restricted the main sample to women who were at least 18 years old, the age by which most Indonesian females in our samples would have experienced menarche. Including younger individuals could introduce selection bias, as those with an earlier menarche (potentially due to prenatal exposure to Ramadan) would have a higher probability of being included in the sample. Such a bias might cause an overestimation of the effects because it would artificially lower the average AAM in the exposed group. In additional analyses that did not rely on the average AAM (see the discussion on Cox regressions below), we used an extended sample that includes all women aged 15–25.

To identify prenatal exposure to Ramadan, exact dates of birth were required for our study. Unlike the IFLS, the Indonesia DHS data only provided the month and year of birth. Thus, we imputed the 15th as the day of birth for all observations in this sample. Heaping in dates of birth was not detected in both samples.

### 2.2. Study Design

#### 2.2.1. Age at Menarche

In both data sources, the age at menarche was self-reported through direct interviews by trained staff [53,54]. The outcome variable AAM is continuous and measured in years. To strengthen the validity of recorded data in the sample, several adjustments were undertaken. First, we included only Muslim women with an age at menarche (AAM) between 9 and 20 years, as data outside this range are likely to be misreported [10,55,56,57]. Particularly in the IFLS sample, when a woman reported an AAM in more than one survey wave, only the first declared AAM was considered, provided that the difference among the stated values did not exceed 1 year. Women who reported an AAM with variation of more than one year across waves are likely to be unsure about their exact age at AAM; so, removing them from the sample increased the precision in our estimation. To further minimize the risk of recall bias, women over 60 years old were excluded from the sample as the AAM is long ago and recall errors may increase by this age [58].

#### 2.2.2. Exposure: Ramadan During Pregnancy

Women were classified into exposure and control groups based on whether their prenatal period overlapped with Ramadan or not. To determine exposure, we calculated 266 days (the average duration of human pregnancy from conception) backwards from the date of birth. This estimated gestational period was then compared against historical Ramadan dates. The women whose calculated 266-day gestation coincided with Ramadan were categorized as exposed, while the control group consisted of Muslims whose gestational period did not overlap with Ramadan. This classification of exposure follows the standard approach in the literature on Ramadan during pregnancy [33]. It does not require information on maternal fasting behavior (see Section 2.3 for details on the quasi-experimental approach).

We further differentiated the exposed group based on the pregnancy trimester during which Ramadan started: trimester 1 covers pregnancy days 1 to 88, trimester 2 spans days 89 to 177, and trimester 3 encompasses days 178 onwards. Note that we placed observations whose conception was calculated to have been within 21 days after the end of a Ramadan into a separate group (“probably-not-exposed”). This avoids noise in the control group since if these individuals were born post-term, they would have experienced Ramadan in early pregnancy [33]. Post-term pregnancies extending more than 21 days beyond the due date are rare [59].

Figure 1 illustrates an example using females born in 1996 and 1997. Diamonds indicate the date of birth. Lines represent the average length of pregnancy (266 days). Circles show the estimated date of conception. A person was considered prenatally exposed to Ramadan 1997 if her time in utero overlapped with Ramadan.

### 2.3. Statistical Methods

We compared the AAM between two groups of Muslim women: those whose prenatal period overlapped with Ramadan, and those who were not in utero during Ramadan. The key advantage of defining exposure based on birth dates is the quasi-random occurrence of Ramadan during a pregnancy. Previous studies on Ramadan during pregnancy in Indonesia showed that the overlap of Ramadan in pregnancy was independent of maternal background characteristics [33,43], in contrast to the subjective maternal decision to fast during pregnancy. At the same time, to the extent that not all women whose pregnancies overlapped with a Ramadan do fast, this implies that offspring whose time in utero overlapped with Ramadan were classified as exposed, even though their mothers did not fast in Ramadan. While this intention-to-treat set-up does not threaten the causal interpretation of our estimates, it leads to a potential bias towards zero in our analyses.

Following the Islamic calendar, Ramadan dates shift 11 days backwards over the Gregorian calendar every year and will be back to the same Gregorian date after a 33-year cycle. Such “shifting-over-the-seasons” characteristics mean that any unobserved season-related confounders in the prenatal phase (e.g., seasonal food availability, likelihood of infectious diseases) are unlikely to bias our results. Therefore, researchers can infer causality by separating Ramadan effects from seasonality effects by controlling for month of birth [33].

Ordinary least squares (OLS) regressions were performed using the two main samples from IFLS and DHS. To gain insights into the relationship between prenatal Ramadan exposure and AAM, we additionally utilized Cox proportional hazard regression on the extended sample that additionally includes females aged 15–17. While OLS looks at the mean difference between the exposed and unexposed groups, Cox estimates the likelihood of experiencing menarche at any given age given the prenatal Ramadan exposure status. Thus, this model focuses on the order and timing of menarche (partial likelihood), rather than requiring the precise AAM (full likelihood approach). This is particularly meaningful regarding DHS data since Muslims younger than 18 years old can still be included in the analysis without leading to selection bias. This allows for a large increase in sample size and thus raises statistical power. For IFLS data, the majority of respondents were at least 18 years old at the interview; so, the Cox model has little added value. The results of Cox regressions are displayed as hazard ratios.

In all analyses, Ramadan exposure was included either as a general dummy (exposed–unexposed), or as trimester-specific sub-categories. Since females living in urban areas tend to experience an earlier menarche and there is a global downward trend of the AAM over time [10], we adjusted the analyses for urban–rural residency, birth year and birth year squared. As previously explained, we also controlled for those women classified as “probably not exposed” and for month of birth. Additionally, survey years were included to account for unobserved confounding factors specific to the timing of data collection. The same set of covariates was used for both OLS and Cox models. Robust standard errors were applied to all analyses.

### 2.4. Sensitivity Checks

In order to address the risk of recall bias [60], we used different subsamples to test the stability of our results: The first subsamples included only women with shorter recall intervals, i.e., max 15 (IFLS) and 6 (DHS) years between declared age at menarche and age at interview. The second subsamples were limited to only younger women, who were thus more likely to correctly recall their AAM at the time of the interview [61], i.e., max 30 (IFLS) and 20 (DHS) years old. Finally, we ran analyses in which we did not adjust for urban–rural residency, and carried out separate analyses separate for urban and for rural females.

## 3. Results

The mean age at menarche in both samples was approximately 13.5 years, aligning with findings from previous research on Indonesia and other low- and middle-income countries [10,62]. When including females aged 15–17 in the DHS sample, the average age at the AAM artificially decreases slightly due to the fact that the mean AAM is based only on those women who already had their menarche. Descriptive statistics for each sample are summarized in Table 1.

Prenatal exposure to Ramadan was not associated with menarcheal onset, neither in the IFLS nor in the DHS sample, independent of the pregnancy trimester during which a Ramadan–pregnancy overlap started (Table 2). Consistent with the OLS estimates (Table 2, columns (1) and (2)), the findings from the Cox analysis (Table 2, column (3)) using DHS-extended data showed no difference in the likelihood of experiencing menarche between the exposed and non-exposed females at any given age. The estimated coefficients were close to zero and the hazard rates for Cox regressions close to one, and the confidence intervals did not fit with the existence of substantial effects.

These findings were stable when reducing the sample to women with shorter recall intervals (Appendix A) and a lower age at the time of interview (Appendix B). The analyses that did not adjust for urbanity, as well as analyses stratified by urban/rural, yielded the same results.

## 4. Discussion

We investigated the effects of prenatal nutrition on the AAM among female Muslims using in utero exposure to Ramadan as treatment. Using two distinct surveys from Indonesia, IFLS and DHS, we found no associations between Ramadan during pregnancy and the AAM, independent of the pregnancy trimester of exposure. Our results are consistent with previous studies on the Dutch famine, in which no association between prenatal undernutrition and menarcheal timing of offspring was found [25,26,49]. Other studies using birthweight as a proxy for maternal nutrition reported mixed results [11]. One key strength of the present study is that our samples included data from a broader range of birth cohorts and more recent time periods, an important consideration given the documented decline in the AAM over time [9].

A few limitations in our study should be noted. Regarding the self-reported AAM, while we accounted for potential recall bias in our sensitivity analyses, the unit of measurement was in years, rather than days or weeks. This adds noise to our dependent variable, and this measurement error would have reduced our statistical power to detect subtler effects of Ramadan exposure during pregnancy. Moreover, we used an intention-to-treat (ITT) framework, i.e., we did not know which pregnant women in our sample did actually fast during Ramadan, but relied on whether or not a pregnancy overlapped with Ramadan. This implies that our ITT estimates are subject to attenuation bias. However, given the high fasting rates among pregnant Indonesian Muslims [34,39,43], and our results being very close to zero, it is unlikely that our conclusions would have been different in a non-ITT set-up. Nevertheless, our method is different from the controlled set-up seen in the work of Nandi et al. [24], in which food supplementation for specific pregnant women was suggested to delay the AAM among offspring. We cannot exclude the possibility that there are heterogeneities in fasting practices between cultural or religious groups or relating to the precise ways in which nutritional composition and dietary quality change during Ramadan. Potentially, for some of such groups, effects of prenatal Ramadan on the AAM that we were unable to uncover in our analyses might exist. Finally, our exposure was defined by the trimester in which Ramadan began, which might not fully capture the variation in exposure intensity within groups. This might introduce a potential source of heterogeneity and should be considered when interpreting the findings.

Furthermore, any effects of nutrition during pregnancy might potentially be mediated by postnatal determinants of AAM, such as childhood or adolescent diets [9,10,11]. An experimental study on rats showed that the interaction between prenatal undernutrition and postnatal high-fat diet led to altered levels of leptin and hypothalamic Kiss1, both of which are crucial for pubertal onset [63]. Unlike in rats, in which pubertal status can be assessed relatively soon after prenatal shocks [18,19,20,21,22], the AAM in humans takes place long after the prenatal exposure, allowing more space for such interactions.

Apart from the pubertal onset, future studies on Ramadan during pregnancy might also look at other outcomes of female reproduction, which occur after puberty starts. For instance, previous studies have found evidence on a link between maternal undernutrition and social indicators of reproductive capacity, such as the timing of the first pregnancy [24,26,49]. However, in contrast to the age at menarche, the age at first childbirth does not reflect the purely biological start of reproduction. This age is socially shaped and might itself be affected by prenatal factors. For instance, earlier Ramadan studies suggested impacts on offspring performance at school and on the labor market [43,64,65,66]. At the same time, education and labor market factors are known to be associated with maternal age at first birth.

A promising avenue for future research into the link between prenatal factors and biological reproductive ages of women is the use of objective clinical and biological markers, which helps overcome challenges arising from the self-reported age at menarche. These include markers such as leptin levels, Tanner staging, hormonal profiles (e.g., estrogen), and intermediate growth outcomes like height-for-age or weight-for-age. The body nutritional status is communicated to the HPG axis through metabolic signals such as leptin, fluctuating levels of which might affect the timing of menarche [67]. An experimental study on lamb documented that poor maternal nutrition during pregnancy affects the offspring leptin level [68]. Furthermore, pubertal status can also be assessed by the changing levels of associated sex hormones (e.g., estrogen) or through clinical scales of Tanner stage [69]. In fact, a few studies on female pubertal onset have looked at clinically measured age at thelarche, the onset of breast development, which is an early indicator of puberty [70]. There is also evidence that prenatal malnutrition can accelerate reproductive maturation, potentially reflecting an evolutionary trade-off in which growth in stature is compromised to prioritize the development of reproductive organs [26]. Investigating the association between in utero exposure to Ramadan and postnatal linear growth could therefore enhance our understanding of the nutritional regulation of reproductive development. While one prior study has explored aspects of this relationship [49], further empirical evidence is needed to clarify the underlying mechanisms and developmental pathways.

## 5. Conclusions

Contributing to the scarce body of empirical evidence on the associations between nutrition during pregnancy and the onset of menarche in the offspring, this study documents that Ramadan during pregnancy is not associated with the AAM. Although some prior research has suggested a link between birthweight and the AAM [9,11] and food supplementation during pregnancy in India was found to delay the AAM [24], our findings, based on Indonesian data, are consistent with previous studies on maternal undernutrition [25,26,49].

In addition to investigating other indicators of female fertility that may be partially socially determined, such as age at first childbirth, future research should consider alternative approaches to studying pubertal onset, such as using clinical markers like leptin levels or sex hormone changes, rather than relying on the self-reported AAM. As a large share of pregnant Muslims fast, this topic is relevant to the large worldwide community of Muslims. At the same time, given the intermittent nature of the Ramadan fast, the relevance of such research may go beyond the health impacts on Muslims as (intermittent) fasting is carried out by many non-Muslims, too.

## Figures and Tables

**Figure 1 nutrients-17-01406-f001:**
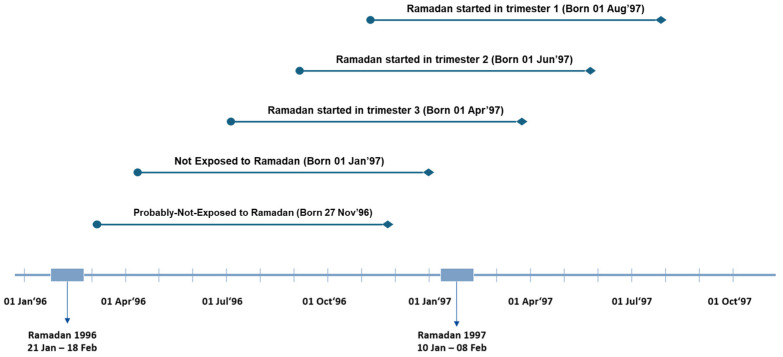
Categories of Ramadan exposure in utero.

**Table 1 nutrients-17-01406-t001:** Characteristics of Indonesian female Muslims from Indonesian DHS (2002–2017) and IFLS (1993–2015).

	IFLS		DHS	
	Main Sample— Aged 18–60 at Observation (n = 8081)		Main Sample— Aged 18–25 at Observation (n = 13,104)	Extended Sample— Aged 15–25 at Observation (n = 24,898)
	Mean (SD)/Share		Mean (SD)/Share	Mean (SD)/Share
Age at menarche (years)	13.66 (1.54)		13.48 (1.41)	13.29 (1.33)
Living in an urban area	58.9%		65.1%	59.4%
Age at the survey (years)	28.25 (6.90)		20.30 (1.90)	18.3 (2.62)
In utero during Ramadan	82.7%		82.6%	82.8%
Ram. started in tri. 1	33.1%		31.3%	31.8%
Ram. started in tri. 2	24.3%		25.5%	26.6%
Ram. started in tri. 3	25.3%		25.8%	24.4%

Abbreviations: IFLS, Indonesian Family Life Survey; SD, standard deviation; Ram., Ramadan; DHS, Indonesian Demographic and Health Surveys.

**Table 2 nutrients-17-01406-t002:** Associations between in utero exposure to Ramadan and age at menarche (in years) among female Muslims.

	IFLS			DHS					
	(1)			(2)			(3)		
	OLS			OLS			Cox		
	(n = 8081)			(n = 13,104)			(n = 24,898)		
Exposure Categories	β	95% CI	*p*	β	95% CI	*p*	HR	95% CI	*p*
In utero during Ramadan	0.025	[−0.082; 0.132]	0.646	−0.005	[−0.086; 0.075]	0.896	0.999	[0.969; 1.030]	0.930
Exposure periods									
Ram. started in tri. 1	0.012	[−0.103; 0.128]	0.833	−0.029	[−0.119; 0.061]	0.529	0.999	[0.966; 1.034]	0.973
Ram. started in tri. 2	0.052	[−0.071; 0.174]	0.407	−0.038	[−0.134; 0.058]	0.436	1.005	[0.970; 1.042]	0.778
Ram. started in tri. 3	0.019	[−0.101; 0.138]	0.760	0.027	[−0.062; 0.116]	0.551	0.995	[0.962; 1.029]	0.771

Notes: The results stem from two separate regressions per column (top panel: exposed vs. not exposed; bottom panel: classification of exposure into different pregnancy phases) that adjusted for “probably-not-exposed” women, birth year, birth year squared, rural–urban living area, month-of-birth, survey wave. Data from the Indonesian Family Life Survey (IFLS) (1993–2015) and the Indonesian Demographic and Health Surveys (DHS) (2002–2017). Column (2) uses the age 18–25, and column (3) the age 15–25 of the DHS sample. Abbreviations: Ram., Ramadan; HR, hazard ratio; CI, confidence interval. β = unstandardized regression coefficient.

## Data Availability

The dataset(s) supporting the conclusions of this article are publicly available at www.dhsprogram.com and www.rand.org. The datasets generated and/or analyzed during this study are available from the corresponding author on request.

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
