# Peer review of "Ramadan During Pregnancy and Offspring Age at Menarche in Indonesia: A Quasi-Experimental Study"

_nutrients, 2025, doi:10.3390/nu17091406_

Round 1

Reviewer 1 Report

Comments and Suggestions for Authors

Congratulation to authors for the very interesting thematic, and the large number of subjects include in the analysis.

Some suggestions to authors:

Abstrat: Line 20 authors write “as intermittent fasting during pregnancy is also widely practiced by non-Muslims”. Can the authors provide more precise information on this data?

Line 20: as intermittent fasting during pregnancy is also 20 widely practiced by non-Muslims.

Line 95: “At the same time, given that Muslim births constitute about 95 31% of the 130 - 140 million babies born worldwide each year”. This is important information, given the magnitude of theses numbers. Suggest to authors to mention the number of pregnant women that fast for Ramadan.  Since previously, in line 80 write that “a very large share decide to fast [26, 27]” it will clear the number if they add a more precise information on how may.

Line 155: Paragraph from line 155 to 159, despite clarifying the methodology, it should be in the introduction chapter.

Line 227: Table 1. “Characteristics of Indonesian female Muslims …” shoul be I results and not in methodology.

Line 248: Table 2: despite being in the legend, the p values ​​are not marked in the table,

Why do the authors fell that the living area (urban) was an important data, if    AAM were not analyzed by living area?

the authors' choice in comparing data by trimester of exposure, places in the same group 1-day exposures vs. 88 exposures, with the same rational for other trimesters. This should also be pointed as a fragility of the study.

Reviewer 2 Report

Comments and Suggestions for Authors

The manuscript explores an original and topical research question: whether prenatal exposure to the Ramadan fast influences AAM in offspring. The study is based on the fetal programming hypothesis and extends the literature by focusing on a mild and intermittent form of maternal nutritional alteration, unlike previous studies that focused on more extreme exposures such as starvation. The use of two large, nationally representative and independent datasets, the robust quasi-experimental design and the application of OLS and Cox regression models are important strengths of the work. The inclusion of sensitivity analyses further supports the reliability of the results.

The presentation is clear and well structured and the manuscript demonstrates a good understanding of the relevant literature. The null results are important and deserve to be published, as they contribute to the ongoing debate on the long-term effects of prenatal nutrition on reproductive health outcomes.

However, an important limitation that deserves further attention is the assumption of maternal nutritional restriction during Ramadan. The study does not measure actual caloric intake, maternal fasting status or changes in diet quality. The authors acknowledge this, but given the centrality of this hypothesis in the study design, a more nuanced discussion would be useful. In particular, the heterogeneity of fasting practices among pregnant women, potential compensatory behaviours during non-fasting hours and cultural/religious variations in fasting adherence could influence the extent of nutritional exposure and should be addressed more explicitly.

Another point to consider is the reliance on self-reported age at menarche, collected in years rather than months. Although the authors conduct appropriate sensitivity analyses to mitigate recall bias, this limitation still affects the precision of the outcome variable and could have contributed to null results if small differences were present.

Finally, although the study appropriately focuses on AAM as a marker of onset of puberty, future research should also explore other biological markers (e.g., leptin levels, Tanner staging, hormonal profiles) or intermediate growth outcomes to better understand the mechanisms linking prenatal exposures to reproductive development.

Round 2

Reviewer 2 Report

Comments and Suggestions for Authors

The answers provided by the authors are comprehensive and well integrated into the manuscript. The points raised in the review have been adequately addressed. I consider the work worthy of publication in its current form.